# Observation of optical gyromagnetic properties in a magneto-plasmonic metamaterial

Weihao Yang[1,4], Qing Liu[2,4], Hanbin Wang[3,4], Yiqin Chen[2,4], Run Yang[1], Shuang Xia[1], Yi Luo[3], Longjiang Deng[1], Jun Qin [1✉], Huigao Duan [2✉] & Lei Bi [1✉]

Metamaterials with artificial optical properties have attracted significant research interest. In particular, artificial magnetic resonances with non-unity permeability tensor at optical frequencies in metamaterials have been reported. However, only non-unity diagonal elements of the permeability tensor have been demonstrated to date. A gyromagnetic permeability tensor with non-zero off-diagonal elements has not been observed at the optical frequencies. Here we report the observation of gyromagnetic properties in the near-infrared wavelength range in a magneto-plasmonic metamaterial. The non-zero off-diagonal permeability tensor element causes the transverse magneto-optical Kerr effect under s-polarized incidence that otherwise vanishes if the permeability tensor is not gyromagnetic. By retrieving the permeability tensor elements from reflection, transmission, and transverse magneto-optical Kerr effect spectra, we show that the effective off-diagonal permeability tensor elements reach $10^{-3}$ level at the resonance wavelength (~900 nm) of the split-ring resonators, which is at least two orders of magnitude higher than magneto-optical materials at the same wavelength. The artificial gyromagnetic permeability is attributed to the change in the local electric field direction modulated by the split-ring resonators. Our study demonstrates the possibility of engineering the permeability and permittivity tensors in metamaterials at arbitrary frequencies, thereby promising a variety of applications of next-generation nonreciprocal photonic devices, magneto-plasmonic sensors, and active metamaterials.

---

[1] National Engineering Center of Electromagnetic Radiation Control Materials, School of Electronic Science and Engineering, University of Electronic Science and Technology of China, Chengdu 610054, China. [2] College of Mechanical and Vehicle Engineering, Hunan University, Changsha 410082, China. [3] Microsystem and Terahertz Research Center, China Academy of Engineering Physics, Chengdu 610200, China. [4] These authors contributed equally: Weihao Yang, Qing Liu, Hanbin Wang, Yiqin Chen. ✉email: qinjun@uestc.edu.cn; duanhg@hnu.edu.cn; bilei@uestc.edu.cn

Metamaterials have attracted considerable research interest in the past two decades because of their artificial electromagnetic properties, facilitating the realization of left-handed materials[1–3], invisible cloaking[4,5], and superlens[6,7]. According to Landau and Lifshitz, the magnetic susceptibility $\chi_m$ vanishes at optical frequencies in conventional materials, that is, $\mu = 1$[8]. Therefore, the realization of optical magnetism with $\mu \neq 1$ and even $\mu < 0$ is of particular interest. Nanostructures including split-ring resonators[9–11], cut-wire pairs[12], dielectric/metal multilayer metamaterials[13,14], dielectric core-metal nanoparticle "satellite" nanostructures[15,16], and all dielectric resonators[17,18] have emerged due to the realization of optical magnetism in recent years. Strong magnetic properties are observed at optical frequencies, leading to the realization of hyperlenses[19], topological transitions[20], and interface-bound plasmonic modes[21,22]. However, so far, only the non-unity diagonal component of the $\mu$ tensor has been demonstrated. The off-diagonal components of $\mu$ are trivial. Can gyromagnetic properties be realized at optical frequencies[23,24]? Answering this question will allow the manipulation of $\varepsilon$ and $\mu$ tensors in metamaterials and facilitate realizing metamaterials with bi-gyrotropic properties at optical frequencies.

Magneto-optical (MO) materials usually show non-zero off-diagonal $\varepsilon$ components at optical frequencies that are also called gyroelectric materials. However, the $\mu$ remains 1 owing to a mismatch of the low Larmor precession frequency of electron spins with the incident electromagnetic field frequency. Therefore, tensorial $\mu$ or even bi-gyrotropic properties (both gyroelectric and gyromagnetic) in MO materials has been pursued for a long time. In the 1950–1960s, gyromagnetic properties were measured in materials such as yttrium iron garnet (YIG) or Fe at optical frequencies. The transverse magneto-optical Kerr effect (TMOKE) under s-polarized incidence was measured that was nontrivial only if the material was gyromagnetic rather than gyroelectric[25–27]. The off-diagonal tensor elements for YIG and Fe were measured to be at most $10^{-5}$,[28] indicating that these materials are not gyromagnetic at optical frequencies. In recent years, with the development of plasmonics and optical frequency metamaterials, magneto-plasmonic devices and MO metamaterials have been developed. These devices show significantly enhanced MO properties in metals, oxides, and 2D materials, thereby promising opportunities for the development of next-generation nanophotonic devices[26–31]. However, almost all MO nanophotonic devices and metamaterials reported so far are limited to the manipulation of the $\varepsilon$ tensor. Optical-frequency gyromagnetic metamaterials are still lacking.

In this study, we report the observation of gyromagnetic properties at near-infrared wavelengths in a magneto-plasmonic metamaterial, which has not been experimentally demonstrated before. Using a classical Au split-ring resonator (SRR) structure on top of an MO thin film Ce-doped YIG (Ce:YIG), we demonstrate s-polarized and p-polarized TMOKE in the hybridized MO-SRR metamaterial. Employing the transfer matrix methods, we demonstrate the off-diagonal element of $\mu$ reaching the $10^{-3}$ level, which is at least two orders of magnitude higher than that of Ce:YIG thin films at ~900 nm wavelength. The microscopic mechanism of the emergent gyromagnetic property is attributed to the local electric/magnetic field direction modulated by the plasmonic nanostructure, underscoring a general strategy of introducing gyromagnetic properties in optical frequency metamaterials. Our study demonstrates the possibility of manipulating the $\varepsilon$ and $\mu$ tensors by hybridizing metamaterials and gyrotropic materials, providing an additional degree of freedom to manipulate electromagnetic waves.

## Results

**Device configuration and operation mechanism.** The hybrid MO-SRR metamaterial is composed of periodic Au SRRs on Ce:YIG/YIG bilayer films, deposited on a silicon substrate, as shown in Fig. 1a, b. The Ce:YIG and YIG thin films are dielectric and transparent MO materials with $n = 2.3$ and $n = 2.1$

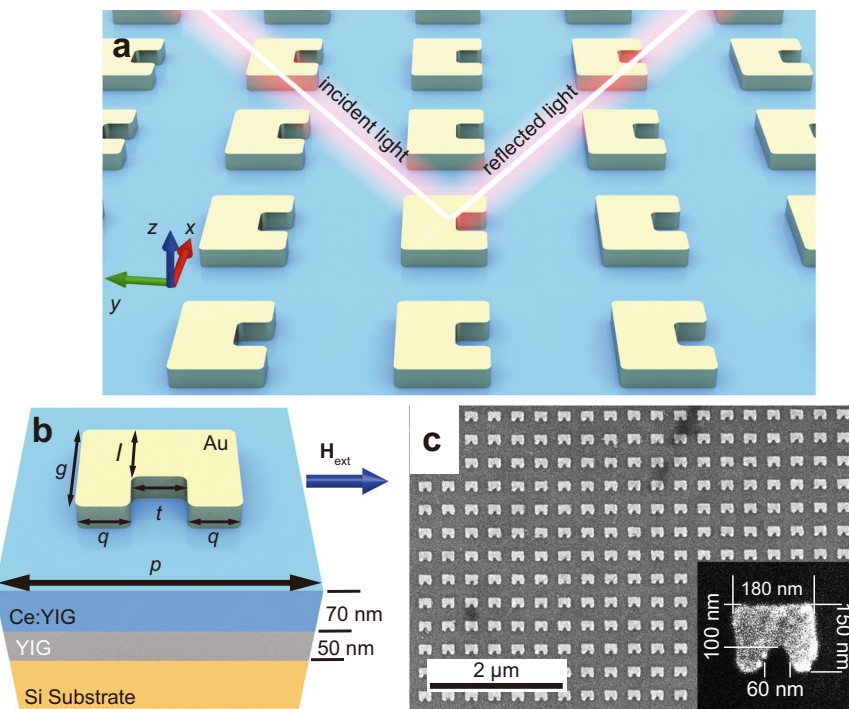

**Fig. 1 Schematic and geometric morphology of the device. a** Schematic diagram of the MO-SRR metamaterial. **b** Schematic of a unit cell of the MO-SRR metamaterial with an Au-SRR on top of the Ce:YIG film (blue), YIG film (gray), and silicon substrate (Orange). The $p$, $g$, $q$, $l$, $t$ are the period of the unit cell, length of vertical arm, width of vertical arm, width of horizontal arm, and width of the gap, respectively. **c** SEM images of the MO-SRR metamaterial.

respectively in the near-infrared wavelength range[25]. The X-ray diffraction spectrum and Faraday rotation hysteresis of the Ce:YIG/YIG films are shown in Supplementary Fig. 1. A linearly polarized light is obliquely incident onto the metamaterial with s- or p-polarization. The applied magnetic field is perpendicular to the plane of incidence (Voigt configuration). Under s-polarized incidence, the electric field is parallel to the SRR gap. The incident electromagnetic wave can couple into both the magnetic and electric resonance modes[29]. Figure 1b shows a schematic diagram of the metamaterial unit cell. The values of each length in the design are $g = 150$ nm, $q = t = 60$ nm, and $l = 105$ nm. The period $p$ is 350 nm, and the thickness of Au is 35 nm. The thicknesses of the Ce:YIG and YIG films are 70 and 50 nm, respectively, measured using cross-sectional scanning electron microscopy (SEM). The top-view SEM image in Fig. 1c shows the morphology of the device with a zoomed-in figure showing one of the SRRs. Compared to the sizes of the design, the experimental results are identical except for the rounded corners of the SRR that is inevitable because of the lift-off process.

**Transmission, reflection spectrum, and mode analysis.** We simulated the transmission and reflection spectra of the device using the commercial software COMSOL Multiphysics based on the finite element method (FEM). In the case of s-polarization with an incident angle of 45°, as shown in the inset of Fig. 2a, we can observe two resonant peaks or dips at 890 and 1350 nm wavelength for the reflection or transmission spectrum, respectively. The measured reflection spectrum shows well consistency with the simulation as the dash line shown in Fig. 2a. The near-field distributions at the interface between the SRR and the

Ce:YIG film in the X–Y plane are shown in Fig. 2b to 2e. First, at 890 nm wavelength, the resonance is induced by the electric dipole oscillating in the horizontal top arm of the SRR[30,31]. Due to the curved shape of the SRR, charge will accumulate at the bottom corners of the two vertical arms, leading to an electric quadrupole mode, as the displacement current ($J_d$) and $E_z$ show in Fig. 2b, c. The directions of the displacement currents are asymmetric in the two vertical arms, which do not form a circulating current with the displacement current in the horizontal top arm, as shown in Fig. 2a, b. In Fig. 2c, The $E_z$ is mainly located at the four corners of the SRR with opposite directions of the adjacent corners (red or blue regions), which is the characteristic of the electric quadrupole mode[32–38]. In fact, similar resonances were excited in a closed ring in previous reports[30]. At the wavelength of 1340 nm, the electric field is concentrated at the corners of the bottom short arms. The displacement currents circulate around the gap, as shown in Fig. 2d, e, leading to an enhanced magnetic field in the gap area and normal to the surface. The resonance at 1340 nm shows the circulating displacement current in the SRR as demonstrated in Fig. 2a. Therefore, this resonance mode corresponds to the magnetic resonance (LC resonance)[39,40]. In the case of p-polarized incidence, the electric field has $y$- and $z$-components perpendicular to the gap. Therefore, only an electric resonance at ~990 nm can be excited, as shown in Fig. 2f. The electric resonance of the two vertical arms can be coupled through the horizontal top arm, leading to red-shifted symmetric and blue-shifted antisymmetric coupling modes. However, the antisymmetric mode is a dark mode that cannot be excited by symmetry protection[41]. This mode can be excited when incident from the orthogonal plane (X–Z plane) (see Supplementary Fig. 2). From the $J_d$ and $E_z$ distributions, we observe the electric field concentrated at the corners of

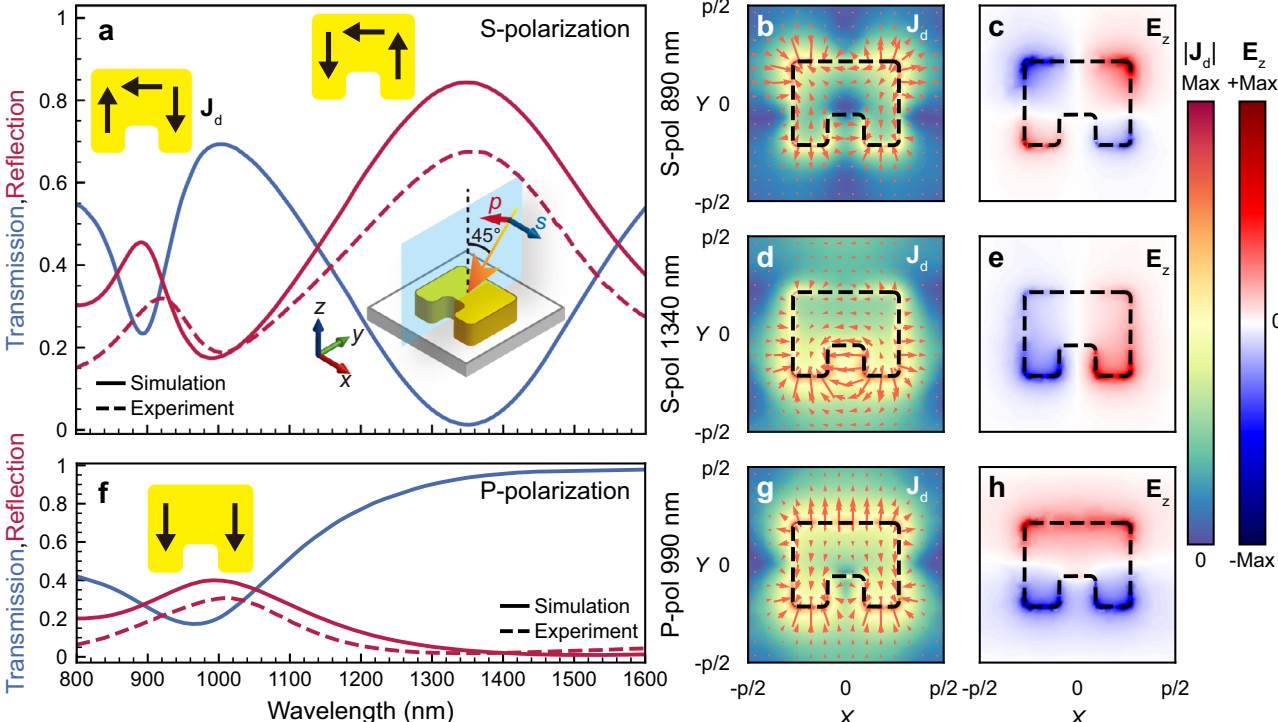

**Fig. 2 Transmission, reflection spectra and mode profiles of the SRRs. a** Transmission and reflection spectra under s-polarized incidence with an incident angle of 45° (dashed line is the measured reflection spectra). The inset shows the volume average of displacement currents $J_d$ distribution in the three arms of the SRR at resonance wavelengths of 890 and 1340 nm. **b** Displacement current and **c** $E_z$ field distribution under s-polarized incidence at 890 nm wavelength. **d** Displacement current and **e** $E_z$ field distribution under s-polarized incidence at 1340 nm wavelength. **f** Transmission and reflection spectra under p-polarized incidence with an incident angle of 45°. The inset shows the volume average of displacement currents distribution in the three arms of the SRR at resonance wavelength of 990 nm **g** Displacement current and **h** $E_z$ field distribution under p-polarized incidence at 990 nm wavelength.

the two vertical arms, corresponding to symmetric coupled electric resonance, as shown in Fig. 2g, h. The measured transmission spectra under s- and p-polarization for normal incidence are shown in Supplementary Fig. 3, which are consistent with the simulation results.

**S-polarized TMOKE.** The permittivity tensor of Ce:YIG and YIG in the Voigt configuration shown in Fig. 1a is

$$\hat{\varepsilon} = \begin{bmatrix} \varepsilon_0 & 0 & 0 \\ 0 & \varepsilon & -j\gamma \\ 0 & j\gamma & \varepsilon \end{bmatrix} \qquad (1)$$

where $\varepsilon_0$ and $\varepsilon$ are the diagonal elements, and $\gamma$ is the off-diagonal elements attributed to the MO effect. It can be directly observed from the above equation that the MO effect would vanish in planar MO films for s-polarized light with electric field along the x-direction. This is because the $\mathbf{E_x}$-induced displacement in the MO material does not couple with $\gamma$. Furthermore, if the s-polarized TMOKE is non-zero, the $\mu$ tensor of the metamaterial must have nontrivial off-diagonal elements owing to the incident $\mathbf{H_y}$ and $\mathbf{H_z}$ field components. This can be better understood from the quantitative relationship of TMOKE and the $\varepsilon$ and $\mu$ tensor elements in a bi-gyrotropic medium, as shown in Supplementary Note 4. We experimentally measured the reflection and TMOKE spectra under s-polarization for both the Ce:YIG/YIG bilayer films and the MO-SRR metamaterial using a spectroscopic ellipsometer (see Methods), as shown in Fig. 3. Figure 3a shows the measured reflection spectra for different incident angles from 45° to 70°. (The maximum range of our ellipsometer is 45°–75°) We can observe two peaks at the wavelengths of ~890 and ~1340 nm, corresponding to the electric quadrupole and magnetic dipole resonance modes, respectively. The displacement current distributions of the two resonance modes are shown in the inset of Fig. 3a. As the incident angle increases from 45° to 70°, the reflectance of both resonances increases. The simulated reflection spectra agree well with the experiments, as shown in

Fig. 3b. To measure the TMOKE spectra, we used a permanent magnet to apply a magnetic field (±3 kOe) in-plane. By switching the poles of the magnet, the reflection spectra under positive and negative magnetic fields were measured. The definition of TMOKE is[42–44]

$$\text{TMOKE} = 2\frac{R_{P/S}(H+) - R_{P/S}(H-)}{R_{P/S}(H+) + R_{P/S}(H-)} \qquad (2)$$

where $R_{P/S}(H\pm)$ is the reflectance of the p-polarized (s-polarized) incidence under a positive or negative applied magnetic field. Remarkably, a clear s-polarized TMOKE reaching $3.0 \times 10^{-3}$ is observed at the electric quadrupole resonance wavelength, as shown in Fig. 3c that significantly contrasts the Ce:YIG thin film exhibiting zero TMOKE signal for any incident angle. The shadow region of the spectra is the standard deviation of five consecutive TMOKE measurements. At the magnetic resonance wavelength of 1340 nm, the amplitude of TMOKE is much weaker than that of electric resonance, which originates from the lower MO effect of the Ce:YIG material (~0.2 times of that at the electric resonance) and the higher reflectance at the magnetic resonance wavelength. We also measured the TMOKE spectrum in the incident angle range of 45°–55°. The TMOKE signal decreases with incident angles that is consistent with the simulation results shown in Fig. 3d. The Ce:YIG thin film did not show s-TMOKE in both the experiments and simulations for different incident angles. According to our simulation, for this device, a maximum s-TMOKE of $2.7 \times 10^{-3}$ may be observed at an incident angle of 30° (see Supplementary Fig. 5).

**P-polarized TMOKE.** According to the equation (16) in Supplementary Note 4, for p-polarization, only the gyrotropic permittivity tensor contribute to the TMOKE. The reflection and TMOKE spectra under p-polarized incidence are shown in Fig. 4. Because the electric field is perpendicular to the gap, only the electric resonance mode is excited at 990 nm, as shown in Fig. 4a. The reflectance decreases with increasing the incident angle owing to a weaker $\mathbf{E_y}$ component that excites the electric resonance. The simulated

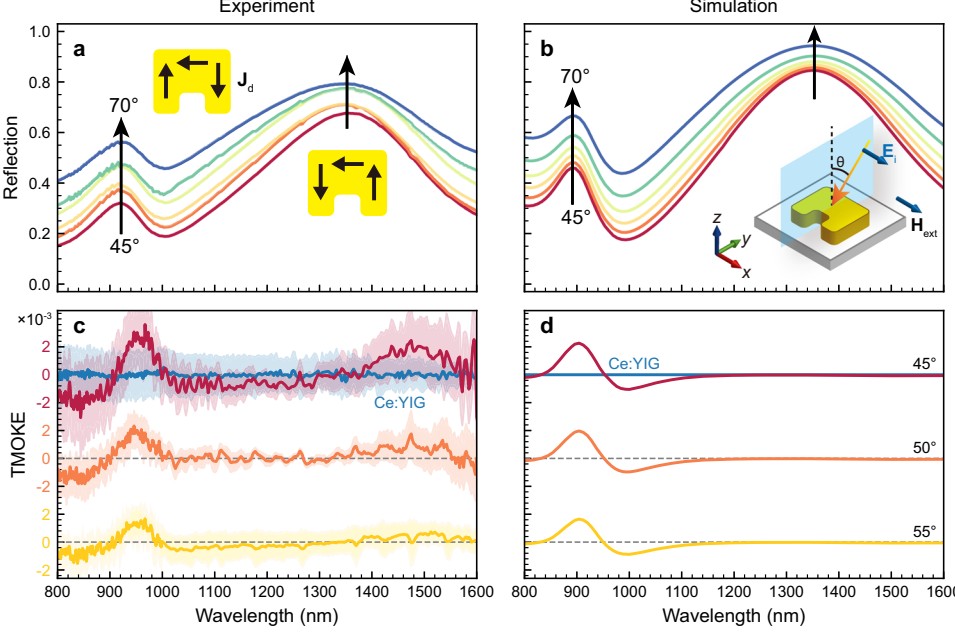

**Fig. 3 Reflection and TMOKE spectra under s-polarized incidence. a** Measured and **b** simulated reflection spectra of the MO-SRR metamaterial for incident angles ranging from 45° to 70° under s-polarized incidence. The inset of **a** shows the displacement current $\mathbf{J_d}$ directions for the electric and magnetic resonances. The inset of **b** shows the schematic of the incident plane, polarization, and direction of applied magnetic field $\mathbf{H_{ext}}$. **c** Measured and **d** simulated TMOKE spectra of the MO-SRR metamaterial and the Ce:YIG thin film for different incident angles under s-polarized incidence.

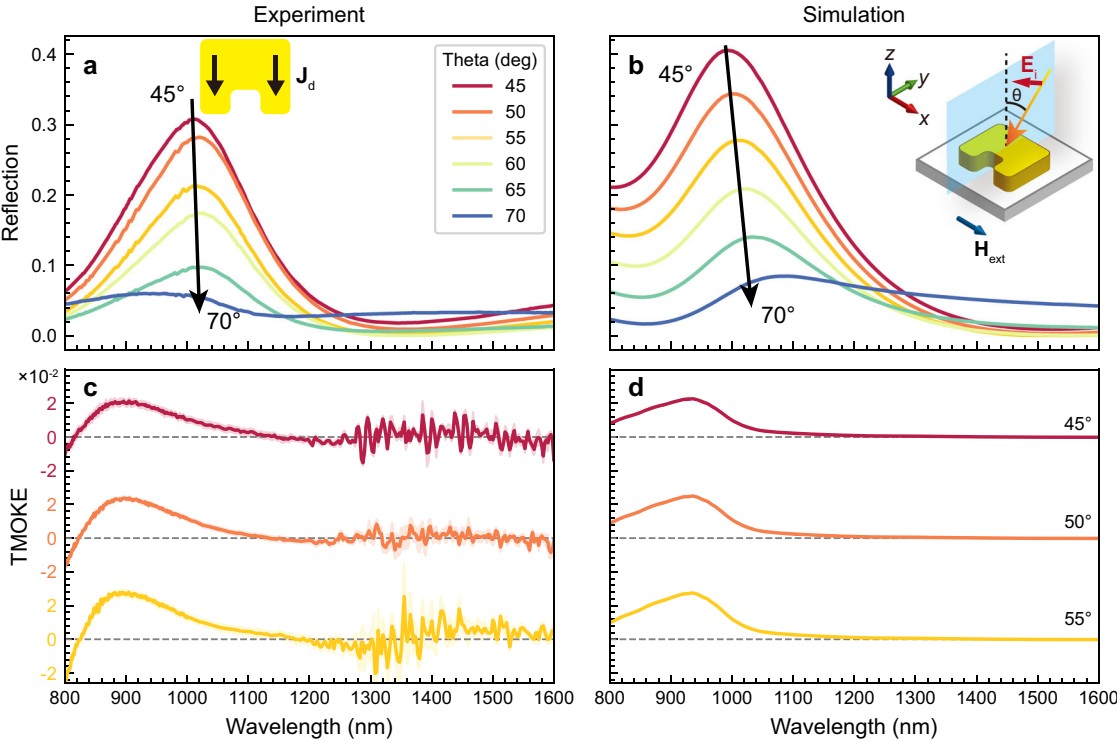

**Fig. 4 Reflection and TMOKE spectra under p-polarization. a** Measured and **b** simulated reflection spectra of the MO-SRR metamaterial and the Ce:YIG thin film for different incident angles under p-polarized incidence. The inset of **a** shows the displacement current $J_d$ directions at resonance wavelength. The inset of **b** shows the schematic of the incident plane, polarization and direction of applied magnetic field $H_{ext}$. **c** Measured and **d** simulated TMOKE spectra of the MO-SRR metamaterial and the Ce:YIG thin film for different incident angles under p-polarized incidence.

reflection spectra agree well with the experiments, as shown in Fig. 4b. For 45° incidence, the TMOKE reaches $2 \times 10^{-2}$ that is 6.7 times higher than the s-polarized TMOKE, as shown in Fig. 4c. The maximum TMOKE shifts to a shorter wavelength relative to the reflection peak, owing to the contributions of the denominator (optical contribution) in Eq. 2. The p-polarized TMOKE of the Ce:YIG film is shown in Supplementary Fig. 6. The larger noise of the spectra ranging from 1300 to 1600 nm is due to the lower signal-to-noise ratio of the ellipsometer at near-infrared wavelengths. According to the definition of TMOKE in Eq. 2, both the increase of $R_{P/S}(H+) - R_{P/S}(H-)$ (MO enhancement) and the decrease in $R_{P/S}(H+) + R_{P/S}(H-)$ (pure optical enhancement) can lead to TMOKE enhancement. Here, the MO contribution is proportional to the amplitude of the electric field inside the MO materials[45,46]. The large enhancement of electric field when exciting the electric resonance mode at ~1000 nm wavelength leads to a higher $R_{P/S}(H+) - R_{P/S}(H-)$, i.e., the MO contribution. For the SRR structure, the reflection shows maximum at the resonant wavelength. Therefore, the pure optical contribution is relatively small. Similar enhancement mechanisms have also been reported in other literatures[45–48]. Because the metamaterial exhibits both s-polarized TMOKE and p-polarized TMOKE in the 800–1000 nm wavelength range, it is called bi-gyrotropic[49,50], which is extremely difficult to achieve in conventional magnetic materials owing to the large difference in the frequencies of the electric dipole transition and magnetic Larmor precession frequency of electrons.

**Effective permittivity and permeability tensors**. To obtain the effective $\varepsilon$ and $\mu$ tensors of the MO-SRR metamaterial, we retrieved the diagonal components of $\varepsilon$ and $\mu$ tensors based on the $2 \times 2$ transfer matrix method (TMM) under glancing incidence for 0 applied magnetic field. The MO-SRR metamaterial is equivalent to a homogeneous layer with an air cladding and the

substrate. According to the standard $2 \times 2$ transfer matrices of multilayers[51], the transmission ($T$) and reflection ($R$) coefficients can be expressed by the permittivity and permeability of the material. By inverting $T$ and $R$, one can obtain the diagonal components of $\varepsilon$ and $\mu$ tensors (see Supplementary Fig. 7).

Thereafter, we treat the MO effects as a perturbation of the $\varepsilon$ and $\mu$ tensors upon applying a magnetic field. We used the $4 \times 4$ TMM to retrieve the off-diagonal elements by fitting the simulated TMOKE spectrum and phase difference of the reflection coefficient between positive and negative magnetic fields. Based on this method, under s-polarization, we retrieved the effective complex refractive indices ($n_s$), permittivity ($\varepsilon_s$) and permeability ($\mu_s$) when the applied magnetic field is zero, as shown in Fig. 5a–c. Figure 5a shows the real and imaginary parts of $n_s$ that implies resonant behavior at both the electric and magnetic resonance wavelengths. For all wavelengths, we observe $\mathrm{Im}(n_s) > 0$, indicating absorption loss of this metamaterial. For $\varepsilon_s$ shown in Fig. 5b, it shows similar resonant behavior to that of $n_s$. In addition, the permittivity at the magnetic resonance is larger than the electric resonance, resulting from the near-zero transmittance at the magnetic resonance wavelength (see Fig. 2a). For $\mu_s$, resonant behavior is also observed at both the electric and magnetic resonance wavelengths[30]. Here, the imaginary parts of both $\varepsilon_s$ and $\mu_s$ show negative signs at around the resonant wavelengths, which have been observed in other metamaterials, and are attributed to the conversion of energy between the electric field and the magnetic field[52–56]. The MO contributions under s-polarization originate from the non-diagonal components of the permeability tensor which is defined as

$$\hat{\mu} = \begin{bmatrix} \mu_0 & 0 & 0 \\ 0 & \mu & -j\kappa \\ 0 & j\kappa & \mu \end{bmatrix} \qquad (3)$$

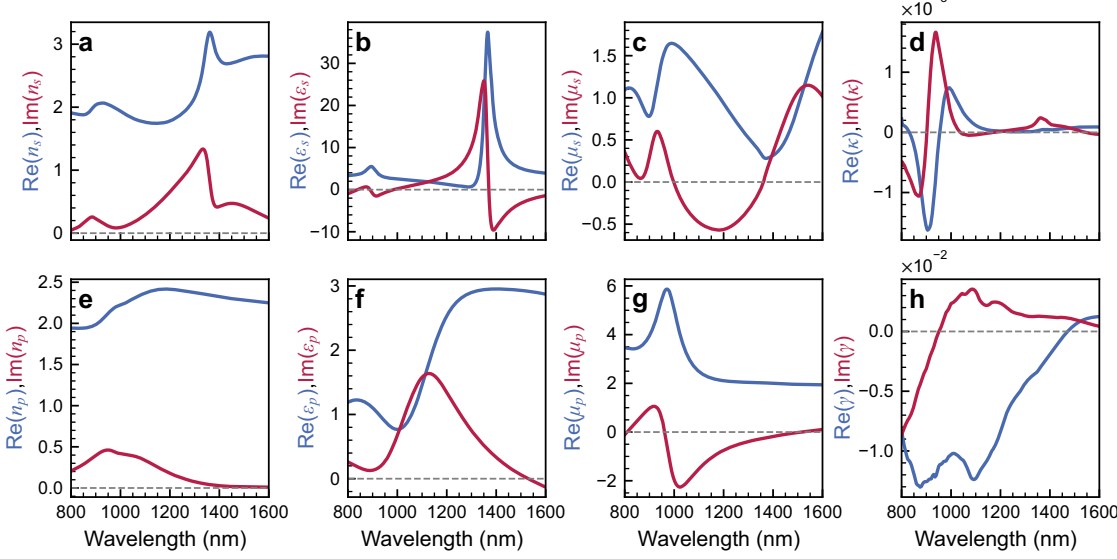

**Fig. 5 Gyrotropic permeability and permittivity tensor of the MO-SRR metamaterials. a** Effective refractive index of the metamaterial under s-polarization. **b** Diagonal components of the complex permittivity under s-polarization. **c** Diagonal components of the complex permeability under s-polarization. **d** Off-diagonal components of the complex permeability tensor. **e** Effective refractive index of the metamaterial under p-polarization. **f** Diagonal components of the complex permittivity under p-polarization. **g** Diagonal components of the complex permeability under p-polarization. **h** Off-diagonal components of the complex permittivity tensor.

where $\mu_0$ is the vacuum permeability, and $\mu$ is the diagonal component of the permeability tensor without an applied magnetic field. $\kappa$ is the MO permeability parameter. Figure 5d shows the retrieved complex $\kappa$ parameters. Both the real and imaginary parts of $\kappa$ can reach $1.6 \times 10^{-3}$ at around the electric resonance wavelength of 890 nm. Similarly, the permittivity tensor can be determined employing the aforementioned method and process under p-polarization, as shown in Fig. 5e–h. The resonant behaviors of $n_p$, $\varepsilon_p$, and $\mu_p$ are observed at the electric resonance wavelength. The non-diagonal components of permittivity $\gamma$ approach $1.3 \times 10^{-2}$. To confirm the correctness of these parameters, we used the transfer matrix method to calculate the transmission, reflection, and TMOKE spectra using the equivalent material parameters that is consistent with the simulated TMOKE spectrum of the MO-SRR. (see Supplementary Fig. 8).

## Discussion

The microscopic origin of the gyromagnetic properties of the MO-SRR metamaterial can be understood based on the mode profile in Figs. 2 and 3a, where the electric field ($E_y$) in both arms of the MO-SRR is orthogonal to the incident electric field direction ($E_x$) upon excitation of the resonance modes. This local change in the electric field direction causes a nontrivial magneto-optical effect according to Eq. 1, resulting in an s-polarized TMOKE. It should be noted that several recent experiments reported the observation of s-polarized TMOKE in magneto-nanophotonic devices[57,58] that share similar microscopic mechanisms of local changes in electric field directions. From a metamaterial perspective, they are all optical gyromagnetic materials. Therefore, a general strategy to introduce optical gyromagnetic properties is to engineer the local electric field direction in magneto-nanophotonic devices, such as magneto-plasmonic metamaterials, MO waveguides, or all dielectric metasurfaces. We have retrieved the permittivity and permeability tensors of recent reported MO metamaterials as shown in Supplementary Fig. 9. Furthermore, this mechanism can be extended to other frequencies. For example, microwave frequency gyroelectric and bi-gyrotropic properties can be observed in magnetic metamaterials (see Supplementary Fig. 10).

The above observation highlights the possibility of engineering the permittivity and permeability tensors in magnetic metamaterials or metasurfaces that enables additional degrees of freedom to control electromagnetic wave propagation. For example, the realization of bi-gyrotropic materials allows controlling the transmission intensity/phase for both the electric and magnetic fields of the incident electromagnetic wave by applying magnetic fields, offering a higher degree of freedom for polarization control in active metasurfaces. The bi-gyrotropic nature of the metasurfaces also enables the independent design of the MO effects, such as TMOKE for p- and s-polarizations that may be used for the design of polarization-independent on-chip MO isolators or circulators[59], nonreciprocal metasurfaces for vectoral magnetic field sensing[60], or biomedical sensing applications[61,62]. Artificial $\varepsilon$ or $\mu$ tensors can also significantly extend the frequency range of magneto-optical properties in conventional magnetic materials that are inherently tied to the band structure or magnetic properties of the material. For example, it is possible to realize strong gyromagnetic/gyroelectric properties in frequencies such as mid-infrared, long-wave infrared, or THz frequencies that are difficult to achieve in conventional magneto-optical materials, facilitating the development of nonreciprocal photonic devices in these frequency ranges.

In summary, we report the observation of optical gyromagnetic properties in an MO-SRR metamaterial in the near-infrared wavelength range. By measuring the TMOKE spectra under s- and p-polarizations, we demonstrate the bi-gyrotropic nature of the magneto-metamaterial. Using the $2 \times 2$ and $4 \times 4$ transfer matrix method, we determine the off-diagonal elements of $\mu$ reaching $1 \times 10^{-3}$, at least two orders of magnitude higher than MO thin-film materials at the same wavelength. A general strategy for local electric/magnetic field vector design in magnetic metamaterials is proposed for the realization of optical gyromagnetic or microwave gyroelectric properties. Our study introduces an additional degree of freedom in metamaterials that paves the way for engineering of the $\varepsilon$ and $\mu$ tensor in advanced nanophotonic structures for nonreciprocal photonic devices, magneto-plasmonic sensors, and active metamaterial applications.

## Methods

**Sample fabrication**. The YIG (50 nm) and Ce:YIG (70 nm) films were deposited on a Si substrate via pulse laser deposition (PLD, TSST). The YIG film was first deposited using a 248 nm KrF excimer laser in a 5 mTorr oxygen ambient at 400 °C. The laser fluence was 3 J/cm$^2$, and the base pressure before the deposition was $1 \times 10^{-6}$ mTorr. After deposition, the film was cooled to room temperature in vacuum, followed by rapid thermal annealing in oxygen ambient (2 Torr) at 850 °C for 3 min, for crystallization. After YIG film crystallization, the Ce:YIG film was deposited in 10 mTorr oxygen ambient at 750 °C and cooled in the same oxygen ambient to room temperature at a rate of 5 °C/min.

The Au SRRs were fabricated via electron beam lithography (EBL, Raith 150-TWO), thermal evaporation, and lift-off process. First, a polymethyl methacrylate (PMMA) resist was spin-coated on the Ce:YIG film at a spin speed of 4000 rpm and baked for 5 min at 180 °C on a hot plate. Thereafter, the nanostructure of the SRR was patterned via EBL with an accelerating voltage of 30 kV and an average dose of 270 µC/cm$^2$. After exposure, the samples were developed in a mixed solution of 3:1 isopropyl alcohol (IPA)/methyl isobutyl ketone for 1 min and rinsed in deionized water for 1 min. Subsequently, a layer of Au was deposited by thermal evaporation (Leybold, UNIVEX250) at room temperature. Finally, the sample was lifted off in an acetone solution with ultrasonic cleaning and rinsed in IPA.

**Optical and magneto-optical simulation**. The optical and magneto-optical responses of the metamaterials were simulated using commercial software based on the FEM (COMSOL MULTIPHYSICS). Periodic boundary conditions were used to account for the periodic structure of the SRR. The polarization and incident angle of light were defined in the incident port. To avoid the effect of the scattered light, a perfectly matched layer was set at the top and bottom layers near the ports. The reflectance and transmittance spectra were calculated using the scattering parameters of COMSOL. For the permittivity of Au, we used the Drude model to fit the permittivity. The refractive index of YIG and permittivity tensor of Ce:YIG were obtained from a previous study[25]. To obtain the TMOKE spectra, the reflectance spectra under positive and negative magnetic fields were simulated. The switching of the magnetic field direction was simulated as the sign change of the off-diagonal elements of the permittivity tensor.

**TMOKE characterization**. The TMOKE spectra were measured using a spectroscopic ellipsometer (J. Woollam RC2) in the wavelength range of 230–1690 nm. The incident angles can change from 45° to 75°. The ellipsometer equipped with two objective lenses can focus on a 100 µm × 200 µm light spot at 45° incidence. The reflectance spectra under s- and p-polarization were obtained through a single measurement. To obtain the TMOKE spectra, a permanent magnet with a 3 kOe magnetic field was used to magnetize the sample in the Voigt geometry. By switching the magnetic field direction, the reflectance spectra were measured under positive and negative magnetic fields. The TMOKE spectra were obtained using Eq. 2.

## Data availability

Most data generated or analyzed during this study are included in this published article or the Supplementary Materials. All data are available from the authors upon reasonable request.

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

## Acknowledgements

The authors appreciate the fruitful discussion with Prof. Ji Zhou of Tsinghua University and Prof. Lei Zhou of Fudan University. The authors are grateful for support by the Ministry of Science and Technology of the People's Republic of China (MOST) (Grant No. 2018YFE0109200 to L.B.), National Natural Science Foundation of China (NSFC) (Grant No. 51972044 to L.B., No. 52021001 to L.D., No. 52102357 to J.Q., No. U1930114 to H.D.), Sichuan Provincial Science and Technology Department (Grant Nos. 2019YFH0154 and 2021YFSY0016 to L.B.), Fundamental Research Funds for the Central Universities (Grant No. ZYGX2020J005 to J.Q.) and the Foundation of CAEP Ultra-precision Machining Technology Key Laboratory (Grant No. ZM18008 to H.W.).

## Author contributions

W.Y. and L.B. conceived the idea. W.Y. performed the device design, metamaterial characterization, and parameters retrieval. R.Y. deposited the magneto-optical thin films. Q.L. and Y.C. fabricated the SRRs nanostructures. S.X. conducted device characterization. H.W. and Y.L. analyzed the data and wrote the paper. J.Q., L.B., H.D., and L.D. supervised the research. All authors contributed to technical discussions and writing the paper.

## Competing interests

The authors declare no competing interests.
