## [Peer Review File · Nature Communications]

Observation of optical gyromagnetic properties in a magneto-plasmonic metamaterialREVIEWER COMMENTS

Reviewer #1 (Remarks to the Author):

The manuscript by W.Yang et al. reports on the observation of gyromagnetic response in the near-infrared wavelength range for a magneto-plasmonic metamaterial. It is found that the non-zero off-diagonal permeability tensor elements appear which give birth to the transverse magneto-optical Kerr effect for s-polarized illumination. This is quite interesting and novel result. Revealing a new physical mechanism for magneto-optical response coming from permeability tensor deserves attention by the optics, magnetism, and metamaterial society. Importantly, this concept is applicable not only for IR, but also to microwave ranges.

The manuscript is well and clearly written. The observed spectra got a proper explanation.

Generally, I would recommend its acceptance to Nature Communications. However, it needs some revision to clarify some points and to amend some issues as proposed below.

Let's start from a general point. How would TMOKE in this metamaterial behave at the gyro-tropic frequency range where both ϵ and μ tensor have already non-diagonal components?

In lines 57-59 authors state: "The transverse magneto-optical Kerr effect (TMOKE) under s-polarized incidence was measured that was nontrivial only if the material was gyromagnetic rather than gyroelectric." Please provide corresponding reference to these papers.

Going further, in lines 59-60: "The off-diagonal tensor elements for YIG and Fe were measured to be at most 10^{-5} at optical frequencies, justifying the argument proposed by Landau and Lifshitz." Actually, Landau and Lifshitz discussed only diagonal part of permeability.

Some explanation about character of the displacement currents shown in Fig.1 and Fig. 3a for ER and MR is needed. At first glance these arrows look unclear.

In Fig.2 distribution of magnetic field demonstrates concentration at the edges of the metal shape for all resonances, however, only the one at 890 nm is identified as electric. Please also explain why one should take care about optical magnetic field for the electric resonance especially.

Line 131 reads: "However, the antisymmetric mode is a dark mode that cannot be excited by symmetry protection". I would ask authors to proof their reasoning and check transmittance for the oblique incidence in the orthogonal plane, i.e. in XZ plane. It will break the symmetry and should make the antisymmetric mode bright.

The authors should explicitly state in the caption of Fig.2 that all presented data is simulated. Since the work is experimental it is important to put experimentally measured transmission spectra in Fig. 2a to compare with calculated one.

Please explain a factor of 2 in Eq.(2).

It should be clarified why for p-polarization only the gyrotropic permittivity tensor contributes to the TMOKE.

Could the authors explain a mechanism of the TMOKE enhancement for p-polarized light at the localized surface plasmon resonance?

In Eq.(3) the permeability tensor should be denoted as μ rather than ϵ .

Fig.5 demonstrates the gyrotropic permeability and permittivity tensor of the MO-SRR metamaterial. It is important to discuss signs of the tensor elements, especially of the imaginary parts. Please also provide similar curves for the effective refractive index of the metamaterial.

Language of the manuscript has to be polished to avoid for example tautology like "The experimental results agree well with the simulation results." etc.

Reviewer #2 (Remarks to the Author):

The work proposed for a review can be attributed to the rapidly developing field of science, which can be called "optical magnetism". It became possible to create artificial structures with subwavelength dimensions with resonances in the optical range, including those with a magnetic moment and magnetic

susceptibility at optical frequencies. The most famous and simplest metasurface with such properties is a periodic set of split-ring resonators made of metal on a dielectric substrate [PRL 95, 203901 (2011)]. Later the TMOKE in bi-gyrotropic media with artificial optical magnetic susceptibility was theoretically considered [i. e. Opt. Lett. 43, 4851 (2018), J. Opt. 21, 03LT01 (2019)]. The combined magnetic dielectric - resonance metasurface structures were also actively studied starting from Ref. 32. The main achievement of the proposed work is the experimental demonstration of TMOKE in the bi-gyrotropic media, which is important but brings a particular advance for a specific community. In my opinion, the work might be suitable for a more specific journal than Nature Comm.

Reviewer #3 (Remarks to the Author):

This manuscript reports an important contribution to magneto-optic gyrotropy. The topic is technologically very significant as the effect is widely used in optical telecommunications for nonreciprocal devices such as optical isolators and circulators. The authors address a particularly interesting question, namely, the contribution to the gyrotropy through the permeability at optical frequencies. It is well known in the field that at these frequencies, the permeability equals to 1 and does not contribute to the Faraday effect. Only the permittivity does. At the same time, researchers have struggled over the years trying to enhance the magneto-optic effect by working with the dielectric permittivity since its magnitude is relatively small. The results that the colleagues report in this manuscript is significant because it opens up a new avenue for the investigation and the fabrication of nonreciprocal devices through the permeability.

The authors present credible and well documented experimental results that show the presence of gyrotropic off-diagonal components in the permeability tensor in the near-infrared regime in split-ring metamaterial resonators. To my knowledge this is the first time such components have ever been observed in this wavelength range. Such off-diagonal (imaginary) components constitute the key to the nonreciprocal magneto-optic effect. So, this is a valuable and noteworthy contribution. Secondly, the authors are able to extract the numerical value of these off-diagonal components through a rigorous analysis of the transmission, reflection and optical modes. So their approach is very thorough.

Because of these reasons I recommend the publication of this manuscript on Nature Communications.

Response to Reviews

Reviewer #1 (Comments to the Author):

The manuscript by W. Yang et al. reports on the observation of gyromagnetic response in the near-infrared wavelength range for a magneto-plasmonic metamaterial. It is found that the non-zero off-diagonal permeability tensor elements appear which give birth to the transverse magneto-optical Kerr effect for s-polarized illumination. This is quite interesting and novel result. Revealing a new physical mechanism for magneto-optical response coming from permeability tensor deserves attention by the optics, magnetism, and metamaterial society. Importantly, this concept is applicable not only for IR, but also to microwave ranges.

The manuscript is well and clearly written. The observed spectra got a proper explanation.

Generally, I would recommend it acceptance to Nature Communications. However, it needs some revision to clarify some points and to amend some issues as proposed below.

Response: We appreciate the reviewer's comments on our work. We now answer the reviewer's questions as follows.

1. Let's start from a general point. How would TMOKE in this metamaterial behave at the bi-gyrotropic frequency range where both ϵ and μ tensor have already non-diagonal components?

Response: Thanks for the comments. In order to clarify the TMOKE in metamaterials with bi-gyrotropic properties, we provide a detailed theoretical analysis starting from the Maxwell's equations. Firstly, let's consider a bi-gyrotropic medium with external magnetic field along the x direction, as shown in Fig. R1a.

Figure R1. Electromagnetic wave propagation in a bi-gyrotropic material (a) Schematic diagram of electromagnetic wave propagation in a bi-gyrotropic material. (b) Schematic diagram of the reflection and refraction of electromagnetic waves at the interface between a bi-gyrotropic material and air. The applied magnetic field H_{ext} is along the x-axis direction.

The permittivity and permeability tensors of such a medium take the form^{r1-3}

$$\bar{\bar{\epsilon}} = \begin{bmatrix} \epsilon_x & 0 & 0 \\ 0 & \epsilon & -j\gamma \\ 0 & j\gamma & \epsilon \end{bmatrix}, \bar{\bar{\mu}} = \begin{bmatrix} \mu_x & 0 & 0 \\ 0 & \mu & -j\kappa \\ 0 & j\kappa & \mu \end{bmatrix} \quad (1)$$

where ϵ and μ are the diagonal components of the permittivity and permeability, γ and κ are the off-diagonal components of permittivity and permeability. The electric and magnetic field vectors can be expressed as $\vec{E} = E_0 e^{-j\vec{k} \cdot \vec{r}}$, $\vec{H} = H_0 e^{-j\vec{k} \cdot \vec{r}}$. Therefore, the Maxwell's equation $\nabla \cdot \vec{B} = 0$, $\nabla \cdot \vec{D} = 0$ can be written as

$$\begin{aligned} \vec{k} \times \vec{E} - k_0 \bar{\bar{\mu}} \vec{H} &= 0 \\ \vec{k} \times \vec{H} + k_0 \bar{\bar{\epsilon}} \vec{E} &= 0 \end{aligned} \quad (2)$$

Where $k_0 = \omega/c$ is the free space wave vector, $\vec{k} = [k_x, k_y, k_z]^T$ is the wave vector in the bi-gyrotropic medium. In the following derivations, we consider the incident plane perpendicular to the external magnetic field, *i.e.* the Voigt geometry. This is also the geometric for measuring the TMOKE.

Consider the case of a homogeneous, bi-gyrotropic medium in Figure R1a, we have $\vec{k} = [0, k \sin \phi, -k \cos \phi]^T$, $\vec{E} = [0, E_y, E_z]^T$, $\vec{H} = [H_x, 0, 0]^T$ for p-polarization and $\vec{k} = [0, k \sin \phi, -k \cos \phi]^T$, $\vec{E} = [E_x, 0, 0]^T$, $\vec{H} = [0, H_y, H_z]^T$ for s-polarization. Here, ϕ is the incident angle with respect to the z axis in the medium, k is the wavevector in the bi-gyrotropic medium, and n_s , n_p are the refractive indices of the medium under s and p-polarization, respectively. Then equation (2) can be factorized into two equations for s and p polarizations:

$$\begin{aligned} \varepsilon_x E_x + \frac{k}{k_0} \cos \phi \cdot H_y + \frac{k}{k_0} \sin \phi \cdot H_z &= 0 \\ \frac{k}{k_0} \cos \phi E_x + \mu H_y - j\kappa H_z &= 0 \quad (\text{s-polarization}) \quad (3) \\ \frac{k}{k_0} \sin \phi E_x + j\kappa H_y + \mu H_z &= 0 \end{aligned}$$

$$\begin{aligned} \mu_x H_x - \frac{k}{k_0} \cos \phi \cdot E_y - \frac{k}{k_0} \sin \phi \cdot E_z &= 0 \\ -\frac{k}{k_0} \cos \phi H_x + \varepsilon E_y - j\gamma E_z &= 0 \quad (\text{p-polarization}) \quad (4) \\ -\frac{k}{k_0} \sin \phi H_x + j\gamma E_y + \varepsilon E_z &= 0 \end{aligned}$$

From equations (3), we see only the μ , κ and ε components are present, while γ is absent from the equations for s-polarization. Whereas only the μ , ε and γ components are present, while κ is absent from the equations for p-polarization. Given the homogeneous equation (3) and (4), they will have non-trivial solutions when the determinant of the coefficients vanishes:

$$\begin{vmatrix} \varepsilon_x & \frac{k}{k_0} \cos \phi & \frac{k}{k_0} \sin \phi \\ \frac{k}{k_0} \cos \phi & \mu & -j\kappa \\ \frac{k}{k_0} \sin \phi & j\kappa & \mu \end{vmatrix} = 0 \quad (\text{s-polarization}) \quad (5)$$

$$\begin{vmatrix} \mu_x & -\frac{k}{k_0} \cos \phi & -\frac{k}{k_0} \sin \phi \\ -\frac{k}{k_0} \cos \phi & \varepsilon & -j\gamma \\ -\frac{k}{k_0} \sin \phi & j\gamma & \varepsilon \end{vmatrix} = 0 \quad (\text{p-polarization}) \quad (6)$$

Thus, we have obtained the generalized formulas for the Cotton-Mouton effect in a bi-gyrotropic medium^{r4}:

$$\begin{aligned} n_s^2 &= \varepsilon_x \mu (1 - Q_m^2) \\ n_p^2 &= \varepsilon \mu_x (1 - Q^2) \end{aligned} \quad (7)$$

Where $Q = \gamma/\varepsilon$, $Q_m = \kappa/\mu$. In addition, for s-polarization, the magnetic field of the wave can be found:

$$\vec{H} = H_0 \begin{bmatrix} 0 \\ \cos \phi + jQ_m \sin \phi \\ \sin \phi - jQ_m \cos \phi \end{bmatrix} \quad (8)$$

where H_0 is magnetic field amplitude. This difference in refractive index for different polarizations provides the possibility to distinguish the gyroelectric and gyromagnetic contributions of bi-gyrotropic materials.

Next, let us consider transmission and reflection of a light wave at the interface between air and a bi-gyrotropic medium. As shown in Fig. R1b, a plane wave is incident from air ($n_1=1$) onto a bi-gyrotropic medium at the incident angle of θ_i , with the electrical vectors perpendicular to the plane of incidence (s wave). The magnetic field of the incident and reflected waves are $\vec{H}_i = H_i [0, \cos \theta_i, \sin \theta_i]^T$, $\vec{H}_r = H_r [0, \cos \theta_r, -\sin \theta_r]^T$, $\theta_r = \theta_i = \theta$. For s-polarization, according to equation (7) $n_2 = n_s = \sqrt{\varepsilon_x \mu (1 - Q_m^2)}$ and Snell's law $\sin \theta_t = (1/n_s) \sin \theta$, the magnetic field of the transmitted wave can be found from wave equation (8):

$$\vec{H}_t = H_t \begin{bmatrix} 0 \\ \cos\theta_t + jQ_m \sin\theta_t \\ \sin\theta_t - jQ_m \cos\theta_t \end{bmatrix} \quad (9)$$

The electric field of the transmitted wave take from $\vec{E} = \frac{1}{k_0} \vec{H} \times \vec{k}$. Considering the continuity of the tangential components of \mathbf{E} and \mathbf{H} at the interface, for s-polarization we have:

$$\begin{aligned} E_{ix} + E_{rx} &= E_{tx}, H_{iy} + H_{ry} = H_{ty} \\ H_i - H_r &= H_t/n_s \\ (H_i + H_r)\cos\theta &= H_t(\cos\theta_t + jQ_m \sin\theta_t) \end{aligned} \quad (10)$$

Hence the formulae for the reflection coefficients in the linear approximation immediately follow:

$$\begin{aligned} r_s &= \frac{E_r}{E_i} = \frac{H_r}{H_i} = \frac{\cos\theta/n_s - \cos\theta_t + jQ_m \sin\theta_t}{\cos\theta/n_s + \cos\theta_t + jQ_m \sin\theta_t} \\ &\approx \frac{\cos\theta - n_s \cos\theta_t}{\cos\theta + n_s \cos\theta_t} + 2jQ_m \frac{\sin\theta \cos\theta}{(\cos\theta + n_s \cos\theta_t)^2} \end{aligned} \quad (11)$$

Similarly, the continuity of the tangential components of \mathbf{E} and \mathbf{H} at the interface, for p-polarization we have:

$$\begin{aligned} E_{iy} + E_{ry} &= E_{ty}, H_{ix} + H_{rx} = H_{tx} \\ E_i - E_r &= n_p E_t \\ (E_i + E_r)\cos\theta &= E_t(\cos\theta_t + jQ \sin\theta_t) \end{aligned} \quad (12)$$

And the reflection coefficient under p-polarization should be expressed as

$$\begin{aligned} r_p &= \frac{E_r}{E_i} = \frac{n_p \cos\theta - \cos\theta_t + jQ \sin\theta_t}{n_p \cos\theta + \cos\theta_t + jQ \sin\theta_t} \\ &\approx \frac{n_p \cos\theta - \cos\theta_t}{n_p \cos\theta + \cos\theta_t} + 2jQ \frac{\sin\theta \cos\theta}{(n_p \cos\theta + \cos\theta_t)^2} \end{aligned} \quad (13)$$

Notice that when Q or Q_m is equal to 0 (a non-magnetic material), equation (11) and equation (13) agree with the Fresnel equations.

Abbreviating equation (11) and equation (13) as

$$\begin{aligned} r_s &= \tilde{r}_s (1 + j\rho_s) \\ r_p &= \tilde{r}_p (1 + j\rho_p) \end{aligned} \quad (14)$$

$$\text{Where } \tilde{r}_s = \frac{\cos\theta - n_s \cos\theta_t}{\cos\theta + n_s \cos\theta_t}, \quad \rho_s = 2jQ_m \frac{\sin\theta \cos\theta}{\cos^2\theta - n_s^2 \cos^2\theta_t}, \quad \tilde{r}_p = \frac{n_p \cos\theta - \cos\theta_t}{n_p \cos\theta + \cos\theta_t},$$

$$\rho_p = 2jQ \frac{\sin\theta \cos\theta}{n_p^2 \cos^2\theta - \cos^2\theta_t}.$$

One can determine the reflectivity ($R=|r|^2$) under positive and negative applied magnetic fields:

$$\begin{aligned} R_s(+H) &= |\tilde{r}_s (1 + j\rho_s)|^2 = |\tilde{r}_s [(1 - \text{Im}\rho_s) + j\text{Re}\rho_s]|^2 \\ &= |\tilde{r}_s|^2 [(1 - \text{Im}\rho_s)^2 + (\text{Re}\rho_s)^2] \\ R_s(-H) &= |\tilde{r}_s (1 - j\rho_s)|^2 = |\tilde{r}_s [(1 + \text{Im}\rho_s) - j\text{Re}\rho_s]|^2 \\ &= |\tilde{r}_s|^2 [(1 + \text{Im}\rho_s)^2 + (\text{Re}\rho_s)^2] \\ R_p(+H) &= |\tilde{r}_p (1 + j\rho_p)|^2 = |\tilde{r}_p [(1 - \text{Im}\rho_p) + j\text{Re}\rho_p]|^2 \\ &= |\tilde{r}_p|^2 [(1 - \text{Im}\rho_p)^2 + (\text{Re}\rho_p)^2] \\ R_p(-H) &= |\tilde{r}_p (1 - j\rho_p)|^2 = |\tilde{r}_p [(1 + \text{Im}\rho_p) - j\text{Re}\rho_p]|^2 \\ &= |\tilde{r}_p|^2 [(1 + \text{Im}\rho_p)^2 + (\text{Re}\rho_p)^2] \end{aligned} \quad (15)$$

Finally, we obtain the TMOKE:

$$\begin{aligned} \delta_s &= 2 \frac{R_s(+H) - R_s(-H)}{R_s(+H) + R_s(-H)} = \frac{-8|\tilde{r}_s|^2 \text{Im}\rho_s}{2|\tilde{r}_s|^2 (1 + |\rho_s|^2)} = -4 \text{Im} \frac{\rho_s}{1 + |\rho_s|^2} \\ \delta_p &= 2 \frac{R_p(+H) - R_p(-H)}{R_p(+H) + R_p(-H)} = \frac{-8|\tilde{r}_p|^2 \text{Im}\rho_p}{2|\tilde{r}_p|^2 (1 + |\rho_p|^2)} = -4 \text{Im} \frac{\rho_p}{1 + |\rho_p|^2} \end{aligned} \quad (16)$$

Ultimately, we can conclude that the TMOKE of s-polarization is only related to the off-diagonal elements of the permeability tensor, and the TMOKE of p-polarization is only related to the off-diagonal elements of the permittivity tensor.

Revision: In the Supporting information, the step-by-step derivation process of TMOKE in a bi-gyrotropic medium is presented in Supplementary Note 4 and marked in red. In lines 165-167, page 9, added “This can be better understood...as shown in Supplementary Note 4.”, and marked in red

2. In lines 57-59 authors state: “The transverse magneto-optical Kerr effect (TMOKE) under s-polarized incidence was measured that was nontrivial only if the material was gyromagnetic rather than gyroelectric.” Please provide corresponding reference to these papers.

Response: We appreciate this advice. We have provided the corresponding references (Ref. 25-27 in the manuscript) to support the argument that the s-TMOKE was nontrivial only if the material was gyromagnetic rather than gyroelectric. This argument is also supported by the derivation in question 1.

Revision: In line 59, page 3, added the [25-27] references, and marked in red.

3. Going further, in lines 59-60: “The off-diagonal tensor elements for YIG and Fe were measured to be at most 10^{-5} at optical frequencies, justifying the argument proposed by Landau and Lifshitz.” Actually, Landau and Lifshitz discussed only diagonal part of permeability.

Response: Thanks for the comments. We have revised this sentence to reflect the fact.

Revision: In lines 59-61, page 3, the sentence is revised to “The off-diagonal tensor elements for YIG and Fe were measured to be at most 10^{-5} at optical frequencies, indicating that these materials are not gyromagnetic at optical frequencies.”, and marked in red.

4. Some explanation about character of the displacement currents shown in Fig.1 and Fig. 3a for ER and MR is needed. At first glance these arrows look unclear.

Response: Thanks for the comments. First of all, it should be pointed out that the direction of displacement currents is opposite to the direction of electric field, since $\vec{J} = -j\omega\epsilon\vec{E}$.¹⁵ In order to clearly explain the displacement currents shown in Fig.1 and Fig. 3a, we have removed the schematic diagram of displacement currents in Fig. 1 and added the illustration of displacement currents distribution in Fig 2a and 2f. They are better understood in the mode analysis. In addition, we changed the electric field distribution to displacement currents in Fig 2b, 2d and 2g. The arrows are corresponding to the volume average of displacement currents distribution in the three arms of the SRR for the modes shown in Fig 2b, 2d and 2g.

Revision: In Fig. 1a, removed the schematic diagram of displacement currents and added the illustration of displacement currents in Fig. 2a and 2f. Changed the electric field distribution to displacement currents in Fig 2b, 2d and 2g. In the caption of Fig. 2, added “The inset shows the volume average of displacement currents distribution in the three arms of the SRR at resonance wavelengths of 890 nm and 1340 nm”, and marked in red.

5. In Fig.2 distribution of magnetic field demonstrates concentration at the edges of the metal shape for all resonances, however, only the one at 890 nm is identified as electric. Please also explain why one should take care about optical magnetic field for the electric resonance especially.

Response: Thanks for the comments. Split ring resonators are anisotropic metamaterials, the best way to see its modes is from the distribution of displacement currents and the out-of-plane components of the electric field (E_z)¹⁶⁻⁸. In the revised manuscript, we replaced the magnetic field distribution at the resonance wavelengths with the distribution of displacement currents and electric fields along z-direction in Fig. 2. At 890 nm, the resonance is induced by the electric dipole oscillating in the horizontal top arm of the SRR^{9, 10}. Due to the curving shape of SRR, the opposite induced charge accumulation will be formed at the bottom corners of the two vertical arms, which will produce an electric quadrupole mode, as the E_z component shown in Fig 2c of manuscript. The E_z is mainly located at the four corners of the SRR with opposite direction of the adjacent corners (red or blue regions), which can be regarded as charge accumulation at these corners. The displacement current in XY plane flows from the red regions to the

blue regions, forming electric quadrupole mode^{r11-14}. At 1340 nm, the E_z component is mainly concentrated on both sides of the SRR gap, and the displacement current flows from red to blue in the metal, forming a circular displacement current corresponding to a magnetic dipole resonance perpendicular to the SRR plane. For incident at 990 nm under p-polarization, E_z component are concentrated on the top and bottom sides, with displacement currents flowing from red to blue along -y direction, forming two electric dipole resonances parallel to each other. More detailed multipole decomposition can be found in previous references [r7]. The classification of the resonance modes agrees with previous studies^{r7, 11, 15, 16}.

Revision: In lines 116-125, page 6, the sentence is revised to “the resonance is induced by ...the characteristic of the quadrupole [32-38]”, and marked in red. In lines 129-131, page 7, added “The resonance at 1340 nm shows a circulating displacement current in the SRR as demonstrated in Fig. 2a.” and marked in red. We replaced the E and H field distribution plots at the resonance wavelengths with the distribution of displacement currents and E_z fields in Fig. 2 and marked in red.

6. Line 131 reads: “However, the antisymmetric mode is a dark mode that cannot be excited by symmetry protection”. I would ask authors to proof their reasoning and check transmittance for the oblique incidence in the orthogonal plane, i.e. in XZ plane. It will break the symmetry and should make the antisymmetric mode bright.

Response: Thanks for the comments. The antisymmetric mode has zero effective electric-dipole moment for p-polarized incidence in the YZ plane, therefore it does not appear. However, when incident from the orthogonal plane, i.e., the XZ plane, this mode can show up due to the broken symmetry. To show this, we simulated and measured the reflectance spectra for oblique incidence in the orthogonal plane (XZ plane). Fig. R2a and R2b show the reflectance spectra for p-polarized incidence in XZ plane with incident angles ranging from 45° to 70° of experiment and simulation. For p-polarization, part of the electric field is along the top horizontal arm direction, therefore both the electric quadrupole and magnetic dipole resonances can be excited. As increasing the incident angle from 45° to 70°, both the reflectivity at the electric quadrupole and magnetic dipole resonances are decreased, due to the decrease of the x-component

electric fields. Both resonances show almost no wavelength shift, because only the x-component electric field contributes to the excitation of the two resonances. The experiment results are consistent with our simulation. However, the experimental magnetic resonance shows a little blue-shifts as increasing the incident angles. This may be resulted from the asymmetric gap area of the fabricated SRR structures, as shown in the inset of Fig. 1c of the manuscript. For s-polarization shown in Fig. R2c and R2d, the electric field is along the two vertical arm direction. For this situation, both the symmetric and antisymmetric electric resonance modes are excited at ~ 900 nm and ~ 1000 nm wavelengths respectively. For oblique incidence, the modes excited at two vertical arms will experience a small time-delay. The near field in the two vertical arms are different, leading to the occurrence of the antisymmetric mode, as shown in Fig. R3, which is also observed in previous studies¹⁷. Therefore, as increasing the incident angles, one electric resonance will split into a red-shifted symmetric mode and a blue-shifted antisymmetric mode, as shown in Fig. R2c. The displacement current directions of the symmetric mode are shown in the inset of the Fig. R2c. In Fig. R2d, we simulated the reflectance spectra under s-polarization, which are consistent with our experiment results. The displacement current directions of the asymmetric mode are shown in the inset of the Fig. R2c.

Figure R2. Reflection spectra under p and s-polarized incidence in the XZ plane **a**, Measured and **b**, Simulated reflection spectra for incident angles ranging from 45° to 70° under p-polarized incidence. The inset of **a** shows the displacement current directions in the SRR. The inset of **b** shows the schematic of the incident plane and polarization. **c**, Measured and **d**, simulated reflection spectra for incident angles ranging from 45° to 70° under s-polarized incidence. The inset of **c** shows the displacement current directions in the SRR.

Figure R3. Reflection spectra under s-polarized incidence in the XZ plane for two incident angles. Simulated reflection spectra for incident angles at **a** 45° and **b** 70° under s-polarized incidence. The inset shows distribution of the E_z component of the surface electric field.

Revision: In lines 138-139, page 7, added “This mode can be excited when incident from the orthogonal plane (XZ plane) (see Supplementary Fig. 2). From the J_d and E_z distributions,” and marked in red. In Supporting information, added Supplementary Note 2, and marked in red.

7. The authors should explicitly state in the caption of Fig. 2 that all presented data is simulated. Since the work is experimental it is important to put experimentally measured transmission spectra in Fig. 2a to compare with calculated one.

Response: Thanks for the comments. We present the simulated transmission and reflection spectra, as well as the experimental reflection spectra for 45° oblique incidence in Fig. 2 in the revised manuscript. The simulation and experimental results show good consistency with each other. Due to the very small sample area (150 μm by 150 μm), it is difficult to measure the transmittance spectrum under oblique incidence angle of 45°. Instead, we measured the transmission spectra of the sample under normal incidence using focusing objectives, as shown in Supplementary Fig. 3. The measured magnetic resonance at 1340 nm wavelength under s-polarized incidence shows broader transmission dip than simulation, which may be attributed to the imperfect sample geometry due to the nanofabrication process. For wavelengths of 1100 nm and shorter, the sample shows low transmission due to the absorption of the thick silicon substrate.

Revision: In Fig. 2a and 2f, added the experimental reflection spectra for 45° oblique incidence. In line 113-114, page 6, added “The measured reflection spectrum shows well consistency with the simulation as the dash line shown in Fig. 2a”, and marked in red. In the caption of Supplementary Fig. 3, added “The low transmission in experiments for 1100 nm and shorter wavelengths is due to the absorption of the thick silicon substrate”, and marked in red.

8. Please explain a factor of 2 in Eq. (2).

Response: Thanks for the comments. The factor 2 is from the definition of TMOKE, which has also been adopted by other literatures^{r18-20}.

Revision: In line 179, page 10, added references [42-44], and marked in red.

9. It should be clarified why for p-polarization only the gyrotropic permittivity tensor contributes to the TMOKE.

Response: Thanks for the comments. According to the equation (16) in the reply of question 1, the TMOKE for p-polarization is only attributed to the gyrotropic tensor of the magnetic material.

Revision: In line 204, page 11, added “According to the equation (16) in Supplementary Note 4”, and marked in red.

10. Could the authors explain a mechanism of the TMOKE enhancement for p-polarized light at the localized surface plasmon resonance?

Response: Thanks for the comments. According to the definition of TMOKE in Eq. (2) of the manuscript, both the increase of $R_{P/S}(H+) - R_{P/S}(H-)$ (MO enhancement) and the decrease of $R_{P/S}(H+) + R_{P/S}(H-)$ (pure optical enhancement) can lead to the TMOKE enhancement. Here, the MO contribution is proportional to the amplitude of the electric field inside the MO materials^{21, 22}. The large enhancement of electric field when exciting the electric resonance mode at ~1000 nm wavelength leads to a higher $R_{P/S}(H+) - R_{P/S}(H-)$, *i.e.* the MO contribution. For the SRR structure, the reflection shows maximum at the resonant wavelength. Therefore, the pure optical contribution is small. Similar enhancement mechanisms have also been reported in other literatures²¹⁻²⁴.

Revision: In lines 217-225, page 12, added “According to the ... reported in other literatures [45-48]”, and marked in red.

11. In Eq. (3) the permeability tensor should be denoted as μ rather than ϵ .

Response: Thank you for correcting this typo. We are sorry for this mistake.

Revision: In Eq. (3), replaced ϵ with μ and marked in red.

12. Fig. 5 demonstrates the gyrotropic permeability and permittivity tensor of the MO-SRR metamaterial. It is important to discuss signs of the tensor elements, especially of the imaginary parts. Please also provide similar curves for the effective refractive index of the metamaterial.

Response: Thanks for the comments. For diagonal elements of equivalent parameters, the curves all show resonant behavior near the corresponding electrical and magnetic resonances. We retrieved the effective refractive index n_s under 0 applied magnetic field, as shown in Fig. 5a and 5e. Resonance peaks were observed at electric and magnetic resonance wavelengths for both $\text{Re}(n_s)$ and $\text{Im}(n_s)$. For all wavelengths, we show $\text{Im}(n_s) > 0$, indicating absorption loss of this metamaterial^{25, 26}. Notice the negative imaginary part of the effective permeability and permittivity at some wavelengths does not imply gain according to the positive $\text{Im}(n_s)$, but rather a transfer of energy between the electric and magnetic fields^{27, 28}. The negative part of effective permittivity and permeability has been observed in different studies using the equivalent material approach which is still widely accepted¹⁷⁻¹⁹. The off-diagonal elements can have positive or negative signs for an MO material, indicating the polarization rotation direction or sign of circular dichroism in this material.

Revision: In Fig. 5a and e, added curves for the retrieved effective refractive index of the metamaterial. In lines 249-253, page 14, added “we retrieved the effective complex refractive indices (n_s)...indicating absorption loss of this metamaterial.” In lines 258-260, page 14, added “Here, the imaginary parts of both ϵ_s and μ_s show...and the magnetic field [52-56]”, and marked in red.

13. Language of the manuscript has to be polished to avoid for example tautology like “The experimental results agree well with the simulation results.” etc.

Response: Thanks for the comments. We have further polished our English language of the manuscript.

Revision: In lines 176, page 9, added “agree well with the experiments”, and removed “The experimental results agree well with the simulation results.” In lines 210, page 12, added “agree well with the experiments, as shown in Fig. 4b”, and removed “The experimental results agree well with the simulation results.”, and marked in red.

Reviewer #2 (Remarks to the Author):

The work proposed for a review can be attributed to the rapidly developing field of science, which can be called "optical magnetism". It became possible to create artificial structures with subwavelength dimensions with resonances in the optical range, including those with a magnetic moment and magnetic susceptibility at optical frequencies. The most famous and simplest metasurface with such properties is a periodic set of split-ring resonators made of metal on a dielectric substrate [PRL 95, 203901 (2005)]. Later the TMOKE in bi-gyrotropic media with artificial optical magnetic susceptibility was theoretically considered [i. e. Opt. Lett. 43, 4851 (2018), J. Opt. 21, 03LT01 (2019)]. The combined magnetic dielectric - resonance metasurface structures were also actively studied starting from Ref. 32. The main achievement of the proposed work is the experimental demonstration of TMOKE in the bi-gyrotropic media, which is important but brings a particular advance for a specific community. In my opinion, the work might be suitable for a more specific journal than Nature Comm.

Response: We would like to thank the reviewer for the comments on the development history of "optical magnetism" in metamaterials. Indeed, "optical magnetism" is a very fast developing field. In such metamaterials, it is now widely recognized that one can achieve non-unity effective permeability by designing photonic nanostructures such as split-ring resonators. However, all the "optical magnetism" metamaterial reported so far shows only non-unity permeability in the diagonal elements, whereas the off-diagonal elements remain zero. Our work dedicates to answer the open question: **Can the off-diagonal elements of μ be non-zero in an optical metamaterial, and how to experimentally construct such a metamaterial?** Compared to previous optical magnetism metamaterial research work, this is a clear step forward by showing that now the off-diagonal part of μ can be engineered. It should be noted that although bi-gyrotropic properties have been recognized important for long time, **it has not been experimentally realized so far, neither in natural materials nor in metamaterials. In fact, there is not even any clear mechanism on how to experimentally realize such a medium.** Although in previous literatures [i.e. Opt. Lett. 43, 4851 (2018), J. Opt. 21, 03LT01 (2019)], the authors pointed out that a variety of new devices including wide-angle superlens, wide angle TMOKE and polarization independent MO devices may be

realized *if bi-gyrotropic medium exists*, they did not show *how to construct* a bi-gyrotropic medium either in theory or in experiments. These previous proposals exactly underscore the importance of exploring the theoretical foundation and experimental methods to realize a bi-gyrotropic metamaterial. This is exactly the contribution of our work. Our work provides clear theoretical derivation, experimental evidence and parameter retrieval process to show a general way to fabricate bi-gyrotropic metamaterials, which could also be applicable in a wide frequency range from the optical to microwave frequencies.

About recent literatures studying magneto-optical dielectric nanophotonic structures, these works focused on the enhancement or modulation of the magneto-optical effects by the waveguide modes or Mie resonance modes in dielectric nanostructures. The ultimate goal in these studies, is to explore all dielectric nanostructures that show significant enhancement of the MO effects for device applications. Whereas in this report, we focus on the development of optical gyromagnetic and bi-gyrotropic metamaterials. The outcome is that we demonstrated a general way to achieve bi-gyrotropic metamaterial in theory, simulation, experiments and at different frequencies. So, the aim, results and contributions of this work are totally different from recent reports on MO all dielectric nanostructures.

In summary, our work is not simply an experimental demonstration of a known concept, but a general way to design and fabricate a new type of metamaterial that gives capability to design the full permittivity and permeability tensor, which is also a step forward for the optical magnetism community. We expect this metamaterial may inspire new designs in future gyrotropic and nonreciprocal electromagnetic wave devices, both in the optical and the microwave frequencies. For example, polarization independent optical isolators and circulators may have new designs now by considering these nanostructures. Therefore, we believe this work makes a novel and important contribution to the optics, magnetism and metamaterial society.

Reviewer #3 (Remarks to the Author):

This manuscript reports an important contribution to magneto-optic gyrotropy. The topic is technologically very significant as the effect is widely used in optical telecommunications for nonreciprocal devices such as optical isolators and circulators. The authors address a particularly

interesting question, namely, the contribution to the gyrotropy through the permeability at optical frequencies. It is well known in the field that at these frequencies, the permeability equals to 1 and does not contribute to the Faraday effect. Only the permittivity does. At the same time, researchers have struggled over the years trying to enhance the magneto-optic effect by working with the dielectric permittivity since its magnitude is relatively small. The results that the colleagues report in this manuscript is significant because it opens up a new avenue for the investigation and the fabrication of nonreciprocal devices through the permeability.

The authors present credible and well documented experimental results that show the presence of gyrotropic off-diagonal components in the permeability tensor in the near-infrared regime in splitting metamaterial resonators. To my knowledge this is the first time such components have ever been observed in this wavelength range. Such off-diagonal (imaginary) components constitute the key to the nonreciprocal magneto-optic effect. So, this is a valuable and noteworthy contribution. Secondly, the authors are able to extract the numerical value of these off-diagonal components through a rigorous analysis of the transmission, reflection and optical modes. So their approach is very thorough.

Because of these reasons I recommend the publication of this manuscript on Nature Communications.

Response: Thanks for the recognition of our work! As the reviewer mentioned, the main contribution of our work is the demonstration of metamaterials with bi-gyrotropic properties in theory and experiments. This property has not been achieved in previously reports. It is particularly important for nonreciprocal photonic device applications, because now both the electric field and the magnetic field of the incident electromagnetic wave show gyromagnetic properties at the same frequency. This will allow new designs for nonreciprocal electromagnetic wave devices, for example to realize polarization independent nonreciprocal photonic devices. It will also inspire subwavelength structure designs in magneto-optical isolators and circulators. We sincerely hope these findings will open a new direction for the development of next generation metamaterials, gyrotropic and nonreciprocal photonic devices.

REFERENCE

- r1. Zvezdin, A. K. & Kotov, V. a. c. A. *Modern magnetooptics and magneto-optical materials*. CRC Press (1997).
- r2. Dmitriev, V. Permeability tensor versus permittivity one in theory of nonreciprocal optical components. *Photon. Nanostr. Fundam. Appl.* **11**, 203-209 (2013).
- r3. Gurevich, A. G. & Melkov, G. A. *Magnetization oscillations and waves*. CRC press (2020).
- r4. Krinchik, G. & Chetkin, M. The problem of determining the dielectric permittivity and magnetic permeability tensors of a medium. *Sov. Phys. JETP* **36**, 1368-1369 (1959).
- r5. Papadakis, G. T., Fleischman, D., Davoyan, A., Yeh, P. & Atwater, H. A. Optical magnetism in planar metamaterial heterostructures. *Nat. Commun.* **9**, 296 (2018).
- r6. Rockstuhl, C., Lederer, F., Etrich, C., Zentgraf, T., Kuhl, J. & Giessen, H. On the reinterpretation of resonances in split-ring-resonators at normal incidence. *Opt. Express* **14**, 8827-8836 (2006).
- r7. Chen, A., Kodigala, A., Lepetit, T. & Kanté, B. Multipoles of even/odd split-ring resonators. *Photonics* **2**, 883-892 (2015).
- r8. Fu, X., *et al.* Mode jumping of split-ring resonator metamaterials controlled by high-permittivity BST and incident electric fields. *Sci Rep* **6**, 31274 (2016).
- r9. Linden, S., Enkrich, C., Wegener, M., Zhou, J. F., Koschny, T. & Soukoulis, C. M. Magnetic response of metamaterials at 100 terahertz. *Science* **306**, 1351-1353 (2004).
- r10. Klein, M., Enkrich, C., Wegener, M., Soukoulis, C. & Linden, S. Single-slit split-ring resonators at optical frequencies: limits of size scaling. *Opt. Lett.* **31**, 1259-1261 (2006).
- r11. Gu, J., *et al.* Terahertz superconductor metamaterial. *Appl. Phys. Lett.* **97**, (2010).
- r12. Okamoto, T., Otsuka, T., Sato, S., Fukuta, T. & Haraguchi, M. Dependence of LC resonance wavelength on size of silver split-ring resonator fabricated by nanosphere lithography. *Opt. Express* **20**, 24059-24067 (2012).
- r13. Chen, S., Reineke, B., Li, G., Zentgraf, T. & Zhang, S. Strong nonlinear optical activity induced by lattice surface modes on plasmonic metasurface. *Nano Lett.* **19**, 6278-6283 (2019).

-
- r14. Ji, Y., Wang, B., Fang, L., Zhao, Q., Xiao, F. & Gan, X. Exciting magnetic dipole mode of split-ring plasmonic nano-resonator by photonic crystal nanocavity. *Materials* **14**, 7330 (2021).
- r15. Liu, N., Guo, H., Fu, L., Schweizer, H., Kaiser, S. & Giessen, H. Electromagnetic resonances in single and double split-ring resonator metamaterials in the near infrared spectral region. *Phys. Stat. Sol. (B)* **244**, 1251-1255 (2007).
- r16. Corrigan, T. D., Kolb, P. W., Sushkov, A. B., Drew, H. D., Schmadel, D. C. & Phaneuf, R. J. Optical plasmonic resonances in split-ring resonator structures: an improved LC model. *Opt. Express* **16**, 19850-19864 (2008).
- r17. Enkrich, C., *et al.* Magnetic metamaterials at telecommunication and visible frequencies. *Phys. Rev. Lett.* **95**, 203901 (2005).
- r18. Kreilkamp, L. E., *et al.* Waveguide-plasmon polaritons enhance transverse magneto-optical Kerr effect. *Phys. Rev. X* **3**, (2013).
- r19. Borovkova, O. V., *et al.* Transverse magneto-optical Kerr effect at narrow optical resonances. *Nanophotonics* **8**, 287-296 (2019).
- r20. Carvalho, W. O. F., Moncada-Villa, E., Oliveira, O. N. & Mejía-Salazar, J. R. Beyond plasmonic enhancement of the transverse magneto-optical Kerr effect with low-loss high-refractive-index nanostructures. *Phys. Rev. B* **103**, (2021).
- r21. Hermann, C., Kosobukin, V., Lampel, G., Peretti, J., Safarov, V. & Bertrand, P. Surface-enhanced magneto-optics in metallic multilayer films. *Phys. Rev. B* **64**, (2001).
- r22. Armelles, G., Cebollada, A., García-Martín, A. & González, M. U. Magnetoplasmonics: combining magnetic and plasmonic functionalities. *Adv. Opt. Mater.* **1**, 10-35 (2013).
- r23. Belotelov, V. I., *et al.* Enhanced magneto-optical effects in magnetoplasmonic crystals. *Nat. Nanotechnol.* **6**, 370-376 (2011).
- r24. Caballero, B., Garcia-Martin, A. & Cuevas, J. C. Hybrid magnetoplasmonic crystals boost the performance of nanohole arrays as plasmonic sensors. *ACS Photonics* **3**, 203-208 (2016).
- r25. Menzel, C., Rockstuhl, C., Paul, T., Lederer, F. & Pertsch, T. Retrieving effective parameters for metamaterials at oblique incidence. *Phys. Rev. B* **77**, 195328 (2008).

-
- r26. Menzel, C., Paul, T., Rockstuhl, C., Pertsch, T., Tretyakov, S. & Lederer, F. Validity of effective material parameters for optical fishnet metamaterials. *Phys. Rev. B* **81**, 035320 (2010).
- r27. Depine, R. A. & Lakhtkia, A. Comment I on "Resonant and antiresonant frequency dependence of the effective parameters of metamaterials". *Phys. Rev. E* **70**, (2004).
- r28. Efros, A. L. Comment II on "Resonant and antiresonant frequency dependence of the effective parameters of metamaterials". *Phys. Rev. E* **70**, (2004).
- r29. Alù, A. First-principles homogenization theory for periodic metamaterials. *Phys. Rev. B* **84**, (2011).

REVIEWERS' COMMENTS

Reviewer #1 (Remarks to the Author):

The authors responded to all my questions and concerns properly and comprehensively. I believe that the manuscript in its current form supplemented with a rather detailed supporting materials could be accepted for Nature Communications.

Reviewer #3 (Remarks to the Author):

Publish in Nature Communications.

Response to Reviews

Reviewer #1 (Comments to the Author):

The authors responded to all my questions and concerns properly and comprehensively. I believe that the manuscript in its current form supplemented with a rather detailed supporting materials could be accepted for Nature Communications.

Reviewer #3 (Comments to the Author):

Publish in Nature Communications.

Response: We are grateful to both Reviewers for the recognition of our work. We also thank the Reviewers for stimulating the improvements of our manuscript.